# Chloroquine Inhibition of Autophagy Enhanced the Anticancer Effects of *Listeria monocytogenes* in Melanoma

**DOI:** 10.3390/microorganisms11020408

**Published:** 2023-02-06

**Authors:** Zuhua Yu, Yingying Zhao, Ke Ding, Lei He, Chengshui Liao, Jing Li, Songbiao Chen, Ke Shang, Jian Chen, Chuan Yu, Chunjie Zhang, Yinju Li, Shaohui Wang, Yanyan Jia

**Affiliations:** 1Luoyang Key Laboratory of Live Carrier Biomaterial and Animal Disease Prevention and Control, Luoyang 471023, China; 2Laboratory of Functional Microbiology and Animal Health, Henan University of Science and Technology, Luoyang 471003, China; 3College of Animal Science and Technology, Henan University of Science and Technology, Luoyang 471003, China; 4Shanghai Veterinary Research Institute, Chinese Academy of Agricultural Sciences, Shanghai 200241, China

**Keywords:** *Listeria monocytogenes*, autophagy, chloroquine, antitumor, melanoma

## Abstract

*Listeria monocytogenes* has been shown to exhibit antitumor effects. However, the mechanism remains unclear. Autophagy is a cellular catabolic process that mediates the degradation of unfolded proteins and damaged organelles in the cytosol, which is a double-edged sword in tumorigenesis and treatment outcome. Tumor cells display lower levels of basal autophagic activity than normal cells. This study examined the role and molecular mechanism of autophagy in the antitumor effects induced by LM, as well as the combined antitumor effect of LM and the autophagy inhibitor chloroquine (CQ). We investigated LM-induced autophagy in B16F10 melanoma cells by real-time PCR, immunofluorescence, Western blotting, and transmission electron microscopy and found that autophagic markers were increased following the infection of tumor cells with LM. The autophagy pathway in B16F10 cells was blocked with the pharmacological autophagy inhibitor chloroquine, which led to a significant increase in intracellular bacterial multiplication in tumor cells. The combination of CQ and LM enhanced LM-mediated cancer cell death and apoptosis compared with LM infection alone. Furthermore, the combination of LM and CQ significantly inhibited tumor growth and prolonged the survival time of mice in vivo, which was associated with the increased colonization and accumulation of LM and induced more cell apoptosis in primary tumors. The data indicated that the inhibition of autophagy by CQ enhanced LM-mediated antitumor activity in vitro and in vivo and provided a novel strategy to improving the anticancer efficacy of bacterial treatment.

## 1. Introduction

Over the past century, substantial progress has been made in the treatment of cancer. However, certain types of tumors remain difficult to treat. Conventional cancer treatments, such as radiation therapy and chemotherapy, place a heavy burden on patients due to their wide ranges of side effects. In addition, despite these therapies, these diseases commonly recur [1]. Therefore, it is necessary to establish novel therapies that can reduce these adverse consequences [2], and microorganisms can be an ideal tool for targeting tumors. The role of bacteria as antitumor agents has been recognized in previous studies [3]. Physicians observed that tumors regressed following accidental *Streptococcus pyogenes* infections in cancer patients. Later, William B. Coley found that cancer patients who developed post-operative bacterial infections were cured of their tumors. Subsequently, a variety of live and genetically modified non-pathogenic or attenuated bacteria are being explored as potential anticancer agents, either to provide direct tumoricidal effects or to deliver tumoricidal molecules [4,5,6].

*Listeria monocytogenes* (*L. monocytogenes*) is a Gram-positive, facultative anaerobic intracellular bacterium that was shown to have antitumor efficacy in 1979 [7]. Once inside cells, some bacteria lyse the phagosome membrane and allow escape into the cytoplasm. LM in the cytosol can deliver foreign antigens to MHC class I and MHC class II pathways and induce specific CD8^+^ and CD4^+^ T cell responses [8]. Previous studies showed that regulatory T cells in tumor tissue were decreased with the action of LM, favoring immune responses that can kill tumor cells [9]. There are studies that have demonstrated that attenuated LM could be a promising cancer vaccine vector [10,11]. Recombinant *L. monocytogenes*-based vaccines expressing tumor-associated antigens (TAAs), including endoglin (CD105) [12], human prostate-specific antigen (PSA) [13], HER-2/neu [14], and HPV16 E7 [15], have been developed for cancer immunotherapy in preclinical and clinical trials. *L. monocytogenes* has been demonstrated to inhibit tumor growth [10,16,17]. However, the mechanism of *L. monocytogenes*-induced tumor cell death has not been well defined.

Autophagy as an intracellular scavenge is a conserved catabolic process that mediates and delivers unfolded proteins and damaged organelles in the cytosol to lysosomes for degradation [18,19]. Autophagy is essential for maintaining homeostasis and can remove unfolded proteins, damaged organs, and intracellular pathogens to prevent cancer development [20]. In contrast, autophagy is a self-defense mechanism that may protect tumor cells from injury caused by nutritional deficiency, ionizing radiation, and chemotherapy during tumor development [19,21]. Hence, autophagy performs a dual role in tumorigenesis and development [22]. Chloroquine (CQ) is an FDA-approved drug used to treat malaria, rheumatoid arthritis, and lupus erythematosus [23]. CQ is a weak base that accumulates in lysosomes, leading to its alkalinization. This blocks the fusion of autophagosomes to lysosomes, resulting in the autophagosomes’ degradation. Therefore, CQ is also being investigated as a novel autophagy inhibitor in clinical trials [24].

The purpose of this study was to investigate the role and mechanism of autophagy in the antitumor effects induced by LM. Specifically, we assessed autophagy induced by LM by Western blotting, real-time quantitative PCR, and ultrastructural analysis and evaluated the antitumor effect induced by the combination of LM and the autophagy inhibitor CQ in vitro and in vivo. Overall, our work will help elucidate an innovative combined therapy to treat cancer.

## 2. Materials and Methods

### 2.1. Bacterial and Cell Lines

The *L. monocytogenes* strain LM-LY03 (serotype 1/2a) was originally isolated from chicken in Luoyang. The LD_50_ of LM-LY03 is 1.1 × 10^6^/mouse; this strain is thus less virulent than that of 10403S. LM was cultured in brain heart leaching broth (BHI) at 37 °C with aeration. B16F10 murine melanoma cells were obtained from the Chinese Academy of Sciences and cultured in 5% CO_2_ in Dulbecco’s Modified Eagle’s Medium (DMEM) with 10% fetal bovine serum (FBS), 100 U/mL penicillin, and 10 µg/mL streptomycin.

### 2.2. Immunoblot Analysis

The effects of LM on autophagy in B16F10 melanoma cells were determined as previously described with some modifications [25]. First, the conversion of LC3-I to lipidated LC3-II in LM-infected cells was determined by Western blotting. Briefly, proteins were extracted with a cell lysis buffer, separated by sodium dodecyl sulfate-polyacrylamide electrophoresis (SDS-PAGE), and transferred onto polyvinylidene fluoride membranes. Then, membranes were blocked and incubated with microtubule-associated protein 1 light chain 3 Ⅱ/Ⅰ(LC3-Ⅱ/Ⅰ) antibody (Cell Signaling Technology) and horseradish peroxidase-conjugated secondary antibody (Cell Signaling Technology) at 4 °C overnight. A GAPDH antibody was used as a control for whole-cell lysates. The images were analyzed by ImageJ 1.43 software.

### 2.3. RT-PCR Analysis

Total RNA was isolated from cell lysates using a TRIzol RNA extraction kit (Invitrogen, Carlsbad, CA, USA) and complementary DNA (cDNA) was synthesized using a PrimeScript RT Reagent Kit (Takara, Dalian, China). Quantitative real-time PCR was performed using a SYBR Premix Ex *Taq* (Takara) and gene-specific primers (Table 1). Relative gene expression was analyzed via the △△CT method. All experiments were performed in triplicate and independently repeated three times.

### 2.4. Analysis of Intracellular Autophagic Vacuoles

B16F10 cells were cultured in 24-well plates. Cells with the GFP-RFP-LC3 adenovirus (HANBIO, Shanghai, China) were transfected at a multiplicity of infection (MOI) of 50 for 36 h and B16F10 cells were infected with LM at an MOI of 100. Finally, the expression of GFP and RFP was observed using an inverted fluorescence microscope (ApoTome.2; Zeiss, Germany). The numbers of GFPC and RFPC puncta (yellow) and RFP puncta (red) were counted.

### 2.5. Transmission Electron Microscopy

Ultrastructural analyses of LM-infected cells were performed by TEM. Briefly, B16F10 cells after 6 h of infection with LM were washed with phosphate-buffered saline (PBS) and treated with 1% osmium tetroxide (Sigma-Aldrich, St. Louis, MO, USA). The cells were embedded in UltraCut and cut into 60 nm sections, followed by staining. Finally, the ultrathin sections were observed with a HITACHI H07650 transmission electron microscope (FEI Ltd., Hillsboro, OR, USA).

### 2.6. Cell Viability Assay

B16F10 cells were pretreated with 20 μM CQ for 6 h and then infected with LM at an MOI of 100 in the presence or absence of CQ for the indicated time. Cell viability was analyzed by CCK-8 assay.

### 2.7. Annexin V-FITC/PI Staining

The LM-induced apoptosis of B16F10 cells was assessed using an Annexin V-FITC/PI Apoptosis Detection Kit (Beyotime Institute of Biotechnology, Shanghai, China). B16F10 cells treated with 20 μM CQ for 6 h were infected with LM and stained with FITC-conjugated Annexin V and PI. Flow cytometry was used to analyze the apoptosis ratio. Experiments were performed in triplicate and independently repeated three times.

### 2.8. Bacterial Intracellular Growth Assays

Bacterial intracellular growth was monitored as described previously [26]. B16F10 cells were first treated with the autophagy inhibitor CQ (20 μM, C6628; Sigma-Aldrich) for 6 h and then infected with LM at an MOI of 100. The cells were lysed at 0, 3, 5, 12, 24, and 48 h after infection. An appropriate dilution of lysate was used for plating and bacterial CFUs were counted to estimate the intracellular titer after incubation overnight at 37 °C. Experiments were performed in triplicate and independently repeated three times.

### 2.9. Ethics Statement

C57BL/6 female mice (7–8-weeks old) (*n* = 6 mice per group) were purchased from Zhengzhou University and were housed in accordance with the protocols approved by the Experimental Animal Center Institutional Committee of the Henan University of Science and Technology. All animal experiments were conducted in accordance with the guidelines of the Humane Treatment of Laboratory Animals.

### 2.10. Tumor Model and Treatment

To establish the tumor model, 5 × 10^5^ B16F10 cells were subcutaneously inoculated into the left flank of female C57BL/6 mice [27]. When the tumor size reached 100 mm^3^ after 1 week, tumor-bearing mice were randomly divided into three groups (*n* = 6 mice per group) and treated with LM alone, LM + CQ, or PBS (as a control). Tumor-bearing mice in the LM and LM + CQ groups were intraperitoneally injected with 1 × 10^5^ CFUs of LM on days 7 and 14 after tumor cell injection. On day 7 after tumor implantation, mice in the LM + CQ group were intraperitoneally injected with 60 mg/kg CQ every other day for 2 weeks [18,28]. Lengths and widths of tumors in each group were measured using a Vernier caliper at regular two-day intervals. Tumor volume was calculated as follows: tumor volume = length × width^2^ × 0.5. The number and dates of deaths of mice were recorded to calculate the survival rate.

### 2.11. TUNEL Staining of Tumor Tissue

To investigate the apoptosis of cancer cells in vivo, tumor tissue sections (10 μm) were prepared according to the manufacturer’s instructions (Beyotime Institute of Biotechnology, Shanghai, China) and stained with a TUNEL BrightGreen Apoptosis Detection Kit (A112-01; Vazyme, China). Images were obtained using a fluorescence microscope (ApoTome.2; Zeiss, Germany). TUNEL-positive cells were counted under the microscope. The apoptosis index was defined by the percentage of TUNEL-positive cells among the total cells of each sample.

### 2.12. Bacterial Distribution in Tumor Tissue

The bacterial colonization of tumor tissues was determined as described previously (Jia et al., 2017). Briefly, C57BL/6 female mice (7–8-weeks-old) were inoculated s.c. with 5 × 10^5^ B16F10 cells at day 0 as described above. Tumor-bearing mice (*n* = 6 per group) were treated with LM alone and LM plus the CQ group on days 7 and 14. The tumor tissues were weighed and homogenized on day 1 and day 3 after each treatment with LM. The bacterial numbers were determined by plating the cell suspensions on BHI agar after incubation overnight at 37 °C and dividing them by the weight of the tissue (CFU/g tissue).

### 2.13. Statistical Analysis

Statistical analysis for the in vitro and in vivo experiments was carried out using GraphPad Prism Software (GraphPad Software Inc., La Jolla, CA, USA). Student’s *t*-test and one-way ANOVA were used for the analysis of comparisons between groups. Three levels of significance (* *p* < 0.05, ** *p* < 0.01, *** *p* < 0.001) were used.

## 3. Results

### 3.1. LM Induced Autophagy in B16F10 Melanoma Cells

The transformation of LC3 (LC3-I) to its autophagosomal-associating form (LC3-II), a classical marker of autophagy, indicated autophagosome formation. To determine whether LM induced autophagy in B16F10 melanoma cells, we first measured the conversion of LC3-I to lipidated LC3-II in LM-infected cells. As seen in Figure 1A, there was obvious transformation of LC3-I to LC3-II in B16F10 cells infected with LM. The ratio of LC3-II to LC3-I in LM-infected cells was significantly higher than in uninfected cells in a time-dependent manner. To determine whether LM induced an increase in autophagy-related gene expression in B16F10 cells, we detected the relative expression of Atg3, Atg5, Beclin-1, and p62 mRNA at 4 h post-infection by real-time PCR. LM infection induced the expression of autophagy-related genes in the B16F10 cells, and the relative expression levels of Atg3, Atg5, and Beclin-1 were higher than in the uninfected group. However, the relative expression of p62 was lower than that of the control group (Figure 1B). To further investigate whether LM induced autophagy, B16F10 cells were infected with the adenovirus GFP-RFP-LC3 followed by infection with LM. A greater accumulation of autophagosomes was observed in LM-infected cells, while few autophagy features were observed in control cells (Figure 1C). To further determine whether LM infection induced autophagosome formation in B16F10 cells, TEM, a standard technique for resolving autophagosomes at the nanometer level, was used to evaluate the accumulation of autophagosomes. In LM-infected cells, partially degraded material could be seen within the autophagic vacuoles, while no similar vesicles were observed in uninfected cells. Therefore, autophagosome formation is consistent with the accumulation of LC3 puncta (Figure 1D).

### 3.2. Pharmaceutical Inhibition of the Autophagy Pathway Enhanced Cell Death Induced by LM In Vitro

Although LM induced autophagy in B16F10 cells, the role of autophagy in the antitumor effects of LM is still unclear. To investigate this, we first blocked the autophagy pathway using CQ in B16F10 cells. As shown in Figure 2A, autophagosomes were observed in LM-treated cells, while few autophagy features were observed in LM + CQ treated cells (Figure 2A). Hence, autophagy could be inhibited by CQ. First, we confirmed that treatment with 20 μM CQ did not affect cell viability. Subsequently, B16F10 cells incubated with CQ for 6 h were infected with LM and cell viability was determined by CCK-8 assay. As seen in Figure 2B, the cell viability of the LM + CQ group was significantly lower than that of the LM control group at 12, 24, 36, 48, 60, 72, 84, and 96 h (*p* < 0.01). Furthermore, we found that the LM + CQ group induced 16.86 ± 0.28% apoptosis, but only 10.21 ± 0.34% in the LM alone group at 60 h post-infection (Figure 2C,D). These findings indicate that inhibition of autophagy via CQ enhanced LM-induced apoptosis.

### 3.3. Autophagy Restricted the Growth of Intracellular LM in B16F10 Cells

To determine the biological role of *Listeria monocytogenes*-induced autophagy, we assessed intracellular bacterial growth following treatment with CQ. Compared with LM treatment alone, the intracellular growth rate of LM was significantly increased after the addition of the autophagy inhibitor CQ (Figure 3). These results indicate that the blockage of LM-mediated autophagy facilitated the intracellular growth of LM in B16F10 cells, suggesting a novel idea of combined therapy to enhance therapeutic effects.

### 3.4. Blockage of Autophagy Potentiated the Antitumor Capacity of LM In Vivo

The antitumor effect of the combined therapy of LM + CQ was assessed in vivo. When the tumor size reached 100 mm^3^ on day 7 after tumor cell inoculation, mice were treated with PBS, LM alone, or LM + CQ. As seen in Figure 4, treatment with LM and CQ significantly inhibited tumor growth and prolonged mouse survival time, indicating that the blockage of autophagy by CQ could enhance the antitumor efficacy of LM in vivo (Figure 4A,B).

### 3.5. Combined Treatment of LM and CQ Enhanced B16F10 Cell Apoptosis In Vivo

To confirm apoptosis following LM treatment in tumor-bearing mice, tumor sections were analyzed with an in situ TUNEL assay. Similarly to the in vitro results, combined treatment with LM and CQ induced more tumor cell apoptosis in vivo (Figure 5A,B). There was a 2.5-fold increase in the number of apoptotic cells in the LM + CQ group compared with that induced by LM alone on day 3 after two intraperitoneal doses.

### 3.6. Inhibition of Autophagy by CQ Enhanced LM Multiplication at the Tumor Site

After confirming that CQ treatment suppressed tumor growth, the number of bacteria in tumor tissues was enumerated. On day 1 and day 3, after two intraperitoneal doses, the mice in each experimental group were euthanized, the tumor tissue was isolated aseptically, and the LM load in the tumor tissue was measured. Bacterial numbers of the combined treatment in tumor tissues were significantly increased compared with LM treatment alone on days 8, 10, 15, and 17 after tumor inocubation (Figure 6). The LM + CQ group showed more LM-targeted tumors compared with the LM group.

## 4. Discussion

In this study, we provided initial evidence that LM could induce autophagy in B16F10 tumor cells and blockage of autophagy enhanced the antitumor effects of LM. Furthermore, the results indicated that the combined treatment of LM + CQ promoted apoptosis and bacterial accumulation in cancer cells, which was associated with the enhancement of the anticancer effects. These observations led us to new insights for combined biological therapy against cancer.

Autophagy has emerged as a conserved innate immune response that restricts the replication of pathogens, including bacteria and viruses, in the cytosol [29]. Studies have shown that LM can induce autophagy in mouse macrophages, and autophagy can inhibit LM escape from vacuoles within 2 h post-infection and enhance intracellular bacterial growth [30,31]. Our results show that LM infection induced the expression of autophagy genes, enhanced the formation of LC3-II, and increased autophagosomes in a time-dependent manner, indicating that autophagy was induced by LM in B16F10 melanoma cells.

The RT-PCR results demonstrate that the relative expression of p62 was lower than that of the control group. Sequestosome 1 (SQSTM1/P62, hereafter referred to as P62) is a multifunctional protein involved in signal transduction, protein degradation, and cell transformation [32]. A marker of the autophagosome, LC3-II, present in the inner membrane of the autophagosome, is degraded together with other cellular constituents by lysosomal proteases. P62 trapped by LC3 is transported selectively into the autophagosome and the inhibition of autophagy results in the accumulation of P62 [33,34]. P62 protein levels are inversely proportional to autophagy activity and thus, P62 serves as a marker of autophagy activity.

The autophagic flux analysis of the action of the adenovirus GFP-RFP-LC3 showed autophagy flux was remarkably inhibited in the LM + CQ group compared with bacterial infection alone. In recent years, CQ has been studied for its potential as an enhancing agent in cancer therapies; it also plays a key role in reversing drug resistance [35]. Accumulating evidence has shown that CQ can be applied in breast cancer metastases, pancreatic cancer, and metastatic carcinoma to inhibit cancer cell growth, promote cell apoptosis, and normalize tumor neo-vessels in the tumor microenvironment [36,37]. Furthermore, CQ has autophagy-independent anticancer properties [28]. In this study, the dosage and treatment schedule of CQ were taken into careful consideration; we finally selected a low dose of chloroquine (60 mg/kg every other day for 2 weeks) in melanoma mouse models and 20 μM of CQ in vitro, which could inhibit autophagy but did not affect tumor size and B16F10 tumor cell viability [18,28]. In this study, the combination of LM and CQ not only increased the growth of intracellular LM, but also resulted in increased rates of cancer cell death and apoptosis in vitro. According to these data, we suggest that the combination of LM and CQ could enhance the cytotoxicity of LM, which is dependent on autophagy, consistent with the anticancer effects of *Salmonella* [18].

Next, we investigated a hypothesis that blocking autophagy would enhance the anticancer effects of LM and promote tumor regression. Similarly to the in vitro results, it was established that autophagy inhibition combined with LM could enhance the ability of cancer cells to promote apoptosis and retard tumor growth. Interestingly, the LM + CQ group showed more LM-targeted tumors than the LM alone group on days 1 and 3 after the first and second treatments, indicating that the autophagy inhibitor CQ enhanced the ability of *L. monocytogenes* to colonize and accumulate in primary tumors, which is associated with antitumor efficacy. The antitumor effect of *L. monocytogenes* was previously attributed to the recruitment of peripheral immune cells to the tumor in tumor-bearing mice [27,38]. Further research is needed to assess whether LM + CQ promote the recruitment of effector immune cells and synergism of the oncolytic effect of *L. monocytogenes*. Furthermore, these conclusions were based on the autophagy inhibitor (CQ) in mice and might not reflect processes in autophagy-deficient mice; therefore, we are now confirming our results in autophagy-deficient mice.

## 5. Conclusions

In this study, we showed that LM induced autophagy in B16F10 melanoma cells. Autophagy helped tumor cells resist LM-induced cytotoxicity and weakened the anticancer ability of LM. The combined treatment of LM and CQ significantly enhanced the anticancer activity of LM, effectively inhibited tumor growth, and prolonged the survival time of mice, which was associated with the increased colonization and accumulation of *L. monocytogenes* and induced more apoptosis in primary tumors via CQ. Furthermore, we found similar results in human esophageal cancer EC9706 cells and a mouse Lewis lung carcinoma LL2 cell line (unpublished data). These findings reveal a novel perspective on autophagy and bacterial biotherapy in cancer and provide proof of concept evidence for the combined therapy of autophagy inhibitors with LM.

## Figures and Tables

**Figure 1 microorganisms-11-00408-f001:**
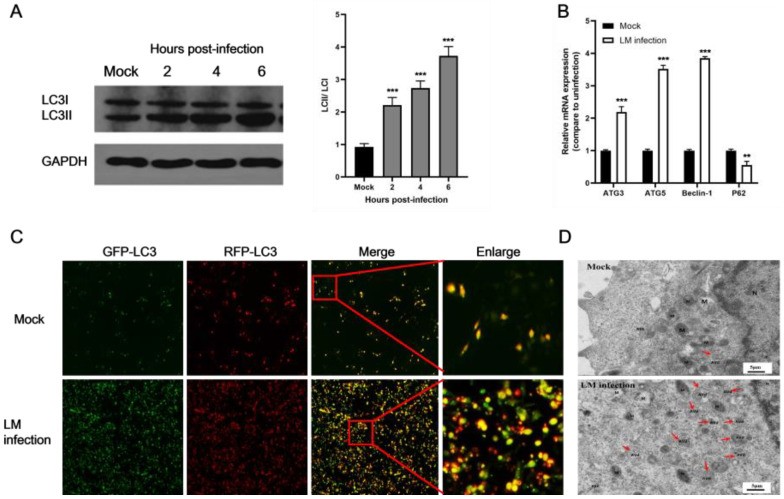
LM induced autophagy in B16F10 murine melanoma cells. (**A**) Autophagy was assessed by Western blotting. B16F10 cells were harvested 0, 2, 4, and 6 h after infection with LM. Expression levels of LC3-I and LC3-II and the conversion of LC3-I to LC3-II were determined by Western blotting. Protein expression ratios of LC3-II/LC3-I were analyzed by GraphPad Prism 5.0. Data are presented as the means  ± SD. (**B**) Quantification of autophagy gene mRNA levels was detected by real-time quantitative PCR. B16F10 cells were harvested at 4 h after infection with LM. The level of autophagy was analyzed by detecting the relative expression levels of Atg3, Atg5, Beclin-l, and p62. Results represent the means ± SD from three independent experiments. (**C**) B16F10 cells were infected with adenovirus with mRFP-GFP-LC3. After 2 h, cells were treated with LM for 1 h and autophagic flux was detected with an inverted fluorescence microscope. Representative images of the fluorescent LC3 puncta are shown. Yellow puncta represent phagophores and autophagosomes, and red puncta represent autolysosomes, Merge (×40); enlarge (×100). (**D**) Ultrastructural analysis by TEM indicated that LM infection induced autophagosome accumulation in B16F10 cells. Red arrows indicate degrading autophagosomes (Avd). M, mitochondria; RER, rough endoplasmic reticulum; N, nucleus. Scale bars, 5 μm (** *p* < 0.01, *** *p* < 0.001).

**Figure 2 microorganisms-11-00408-f002:**
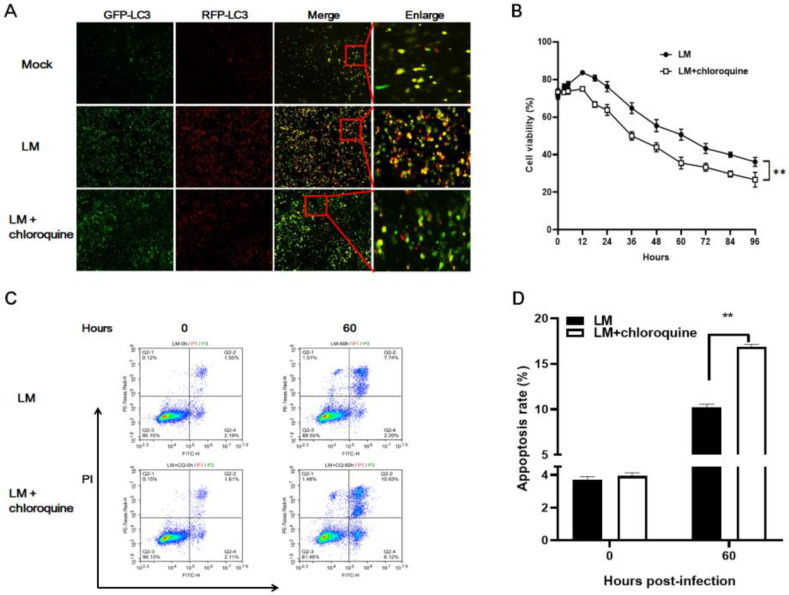
Inhibition of autophagy induced more B16F10 cell death than LM infection alone. (**A**) B16F10 cells were infected with adenovirus expressing GFP-RFP-LC3 for 36 h followed by treatment with LM, CQ, or LM + CQ. Autophagosomes were observed by fluorescence microscopy. Representative images of fluorescent LC3 puncta are shown. Yellow puncta represent phagophores and autophagosomes, and red puncta represent autolysosomes. GFP-LC3, RFP-LC3, merge (magnification, ×40), and enlarge (×100). (**B**) B16F10 cells were infected with LM at an MOI of 100 in the presence or absence of 20 μM CQ for the indicated time. Cell viability was analyzed by CCK-8 assay. Data are presented as the means ± SD. (**C**,**D**) B16F10 cells were infected with LM at an MOI of 100 by the action of 20 μM CQ. Cell death was determined by flow cytometry. Data are presented as the means ± SD (** *p* < 0.01).

**Figure 3 microorganisms-11-00408-f003:**
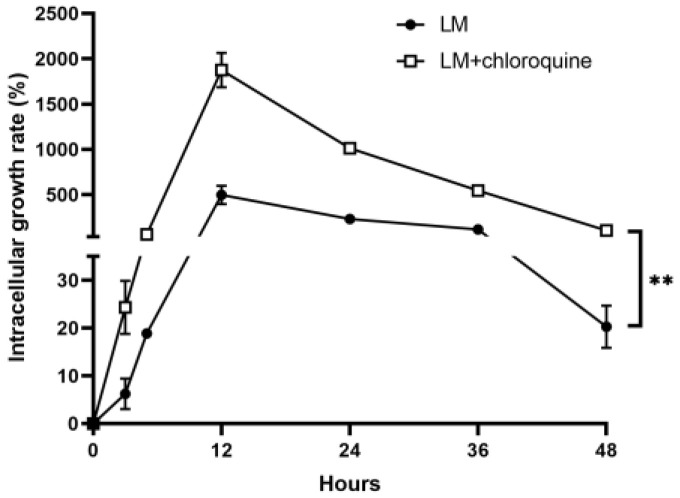
Autophagy restricted the growth of LM in B16F10 cells. Cells were treated with the autophagy inhibitor CQ for 6 h, infected with LM at an MOI of 100 and lysed at 0, 3, 5, 12, 24, and 48 h after infection. Intracellular bacterial measurements were performed as described in the Materials and Methods. Data shown as the means ± SD represent the number of bacteria that invaded cells from three independent experiments, ** *p* < 0.01.

**Figure 4 microorganisms-11-00408-f004:**
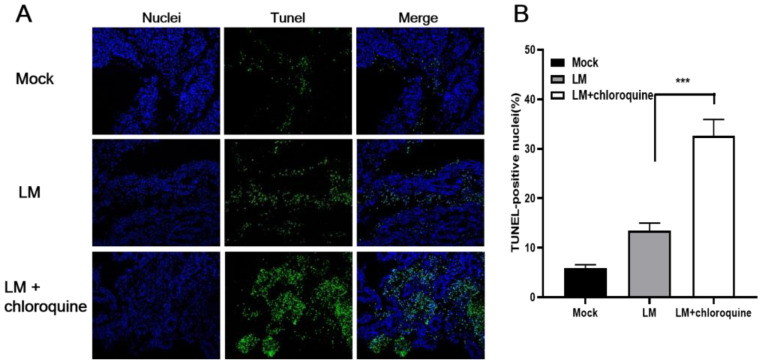
Synergistic inhibition of tumor growth in tumor-bearing mice with combined treatment with LM + CQ. (**A**) Survival rates of mice bearing B16F10 melanomas were analyzed by Kaplan–Meier survival curves. Survival curves were compared by the log-rank test. (**B**) Tumor-bearing mice (*n* = 6 for each group) were treated with LM, PBS, or LM + CQ. Tumor volumes in the different groups were compared. All data are presented as the means ± SD. *** *p* < 0.001.

**Figure 5 microorganisms-11-00408-f005:**
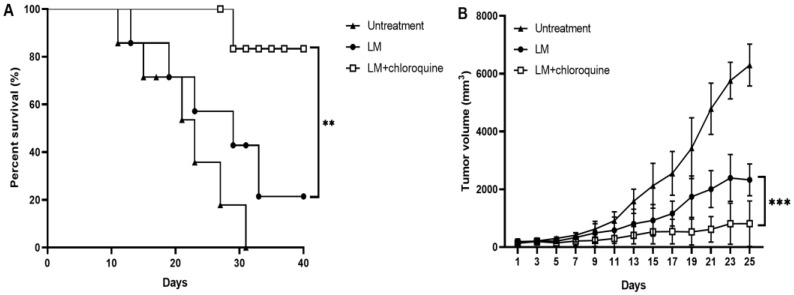
Combined therapy of LM and CQ enhanced cell apoptosis. (**A**) Tumor tissues were excised on day 3 after two intraperitoneal doses and tumor sections were analyzed with an in situ TUNEL assay. TUNEL staining showing apoptosis in mice treated with PBS, LM, or LM + CQ. (**B**) TUNEL-positive cells were counted and the TUNEL index was calculated based on the number of total nuclei and the number of cells with green nuclei in each section. All data are presented as the means ± SD. ** *p* < 0.01, *** *p* < 0.001.

**Figure 6 microorganisms-11-00408-f006:**
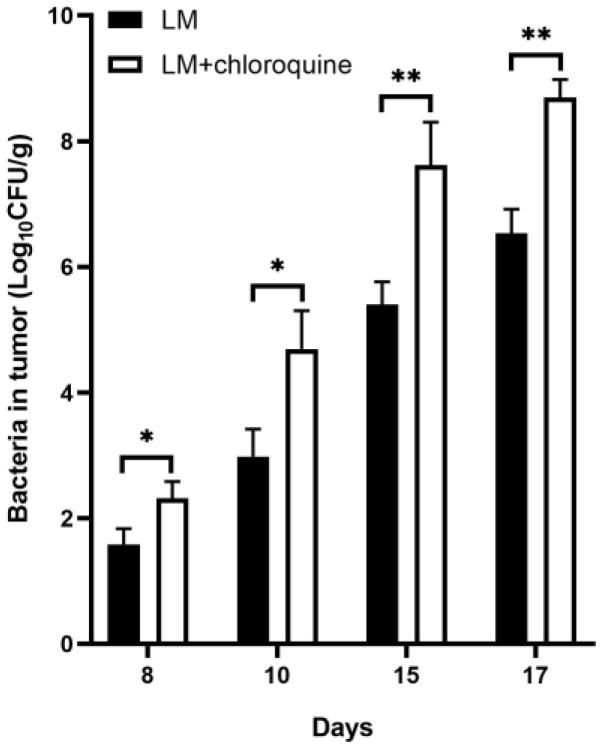
Combined therapy of LM + CQ enhanced bacterial tumor-targeting ability. LM was intraperitoneally infected with a tumor-bearing mice model (*n* = 6 for each group). On day 1 and day 3, after two intraperitoneal doses, the mice in each experimental group were euthanized and the bacterial load in the tumor tissue was determined. The y-axis represents the logarithm of viable bacterial CFU to base 10 in the tumors (* *p* < 0.05, ** *p* < 0.01).

**Table 1 microorganisms-11-00408-t001:** Autophagy-related genes and their corresponding primers used in this study.

Target Gene	Primer	Sequence (5′ to 3′)	Product Size (bp)
β-actin	β-actin-F	CCACGAAACTACCTTCAACTCC	132
	β-actin-R	GTGATCTCCTTCTGCATCCTGT	
Atg3	ATG3-F	CTGGCGGTGAAGATGCTATT	201
	ATG3-R	GTGGCAGATGAGGGTGATTT	
Atg5	ATG5-F	TGGGCCATCAATCGGAAACTC	129
	ATG5-R	TGCAGCCACAGGACGAAACAG	
Beclin-1	Beclin-1-F	AATGACTTTTTTCCTTAGGGGG	142
	Beclin-1-R	GTGGCTTTTGTGGATTTTTTCT	
p62	P62-F	GCACACCAAGCTCGCATTC	124
	P62-R	ACCCGAAGTGTCCGTGTTTC	

## Data Availability

Not applicable.

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
