# Peer review of "Chloroquine Inhibition of Autophagy Enhanced the Anticancer Effects of Listeria monocytogenes in Melanoma"

_microorganisms, 2023, doi:10.3390/microorganisms11020408_

Round 1

Reviewer 1 Report

The manuscript reports very interesting observations that might pave the way to improved bacterial-based anticancer therapies. 

Unfortunately, some flaws are present, the major one being the lack of "negative" controls of treatment with chloroquine alone.

Other points are reported as notes in the attached pdf.

Reviewer 2 Report

The article “Chloroquine inhibition of autophagy enhanced the anticancer effects of Listeria monocytogenes in melanoma” brings more knowledge and can be published in Microorganism journal, but after checking the text regarding English.

Author Response

Dear Reviewer:

Thank you for your comments concerning our manuscript entitled "Chloroquine inhibition of autophagy enhanced the anticancer effects of Listeria monocytogenes in melanoma" microorganisms-2147049.Those comments are very helpful and all valuable for revising and improving the paper. We have revised our manuscript according to your suggestion, and we hope that the correction will meet with approval. Once again, thank you very much for the comments and kind advice.

Sincerely yours,

Zuhua Yu

Reviewer 3 Report

This study showed LM and CQ inhibited tumor growth and prolonged the survival time compared to LM alone. The following problems should be concerned.

of mice, which was associated with increased colonization and accumulation of L.

1.        In section 3.2, we did not see Figure 2.

2.        Figure 5A, B were missing.

3.        Line 28:Tumor cells display lower levels of basal autophagic activity than normal cells.

1)Is that in contrast to the lines 72-73 "In contrast, high levels of autophagy are induced in tumor cells to promote tumor development in hypoxic tumor microenvironments? Hypoxia increases the level of autophagy in tumor cells, however, the abstract states that the basal level of autophagy in tumor cells is lower than that in normal cells;

2)Adding the corresponding references for question 1.

4.        Line 81-82“L. monocytogenes has been demonstrated to inhibit tumor growth. However, the mechanism of L. monocytogenes-induced tumor cell death has not been well defined. It would be better to move it to after the second paragraph of the background?

5.        Line 108/116: to determine..., to further investigate...” Such statements are often used to describe the process of outcome inquiry rather than in the Materials and Methods section. Just introduce the method directly.

6.        Line 122 has a similar problem to lines 108/116: “Transmission electron microscopy (TEM) is a standard technique for resolution of autophagosomes in the nanometer range.”An overall evaluation of TEM is not quite appropriate here and is recommended to be deleted and just describe the specific method. If the importance of TEM is warranted to be highlighted, line 205-207 can be changed to:TEM, a standard technique for resolving autophagosomes at the nanometer level, was used to evaluate the accumulation of autophagosomes. In LM-infected cells, partially degraded material could be seen within the autophagic vacuoles, while no similar vesicles were observed in uninfected cells.

7.        Why emphasize the sex of the model mice with female mice? Whether the stability of the results of this experiment is influenced by the sex of the mice, adding the corresponding references.

8.        Since the colonization and cloning of LM bacteria in tumor cells increased after inhibition of autophagy, does promoting higher levels of autophagy in tumor cells lead to lower colonization and cloning of LM in tumor cells?

9.  In the Discussion section, the limitations of this experiment should be briefly described.

10.  Line 376:Is the term "between" accurate?

11.  Check if there are any relevant studies in the recent years and update the references.

Round 2

Reviewer 1 Report

The authors addressed the reviewer's concerns.